# Building non-motorized accessible communities for heterogeneous demand: A facility location and network design problem

**Stefano Starita** *, **Pinnaree Tea-Makorn** , **Pavitra Jindahra**

Data Analytics Center (DAC), Sasin School of Management, Chulalongkorn University, Bangkok, Thailand

* stefano.starita@sasin.edu

## Abstract

Improving non-motorized accessibility is essential to enable access to vital services such as education and healthcare. Road networks are often inadequately designed for non-motorized accessibility, and service facilities are not always strategically located to reach the majority of users. Furthermore, design decisions are rarely made with consideration for the diverse needs and walking abilities of a heterogeneous population. Enhancing infrastructure by locating new service facilities and creating accessible paths involves significant investments, requiring a systematic approach to utilize limited resources efficiently. We propose an optimization model to identify optimal locations for new facilities and accessible road segments to maximize non-motorized accessibility to a range of services, taking into account users with different walking abilities. A randomized greedy algorithm is developed to tackle the complex network design and facility location problem. A case study in the Huai Kwang district of Bangkok is discussed to illustrate how this methodology can support strategic decisions to enhance non-motorized accessibility.

**Data Availability Statement:** Data is in the form of shapefiles, uploaded on a google drive folder accessible from everyone having the link: https://drive.google.com/drive/folders/

## Introduction

Accessibility to critical services is not only a fundamental human right but also a social justice issue. Increasing access to essential services such as education and healthcare can help reduce social inequalities, promote social inclusion, and enhance the overall quality of life. In this paper, we focus on non-motorized accessibility which refers to transportation modes such as walking, cycling and small-wheeled transport, including wheelchairs.

Enhancing accessibility is crucial for both health and socio-economic reasons. Reduced mobility is associated with future disabilities. Studies by [1, 2] have shown that the inability to walk a quarter of a mile leads to higher annual healthcare costs. Given the consistent worsening of income inequality [3] and the fast pace of ageing [4], car transportation is likely to become a less viable option for a growing share of the population. Moreover, road traffic is one of the leading causes of pollution. The European Environment Agency estimates that every year 110,000 disability adjusted life years are lost in Europe in people under the age of 18 [5]. This figure is likely worse in South East Asia, where people are frequently exposed to unhealthy levels of PM 2.5 [6].

1T8UiR7N9ZOOb5jcYGf0S57dnmEhcrlEE?usp=drive_link.

**Funding:** This research is supported by Rathchadapiseksompotch Fund Chulalongkorn University. The funders had no role in study design, data collection and analysis, decision to publish, or preparation of the manuscript.

**Competing interests:** The authors have declared that no competing interests exist.

Accessibility issues can be equally observed in high and low income countries. In the US, for example, less than 3% of workers commute by walking [7] and the infrastructure investments have historically focused on cars, rather than public transportation. This has resulted in car-centric suburbs, restricting the access to services for children and elderly. In developing countries, rapid urbanizations have led to high car-dependency as policies and infrastructure are often designed to increase motorization [8].

Over the years, car-centric urban planning has resulted in the development of infrastructure riddled with architectural barriers, which hinder pedestrian movement and limit accessibility for those with mobility challenges.

Transforming and adapting infrastructure with a focus on long-term demographic changes is essential to meet the needs of the most vulnerable populations. Failure to do so will lead to negative socio-economic consequences, with segments of the population unable to access basic services. Decision-making should be done accounting for different demographic groups with different abilities and needs. For example, a facility may be easily accessible to healthy individuals, while remaining out of reach for the elderly or those with mobility impairments. Developing infrastructure without accounting for heterogeneous demand needs risks alienating a significant portion of users.

In this work, we aim to develop a systematic approach to improve accessibility by enhancing the non-motorized accessible network and locating new service facilities considering different demand groups and service types. Given that the budget for enhancing accessibility is often limited, it is critical to devise systematic tools that can help allocate resources effectively.

The first issue to address is how to define and measure accessibility. Accessibility measurement is a complex, context-dependent task that varies significantly depending on the purpose of a study. Several methods have been proposed in the literature. For instance, distance-based measures rely on the travel distance or time required to reach a destination [9–11]. Gravity-based measures consider the attractiveness of a destination and the distance or travel time required to reach it [12, 13]. Network-based measures take into account the structure and connectivity of the transportation network, and multimodal measures consider multiple modes of transportation [14].

The scope of the measurements is also a factor. Some paper focus on regional accessibility [15, 16], while others narrow the focus to local communities [17]. This sparked an interesting debate on whether regional or larger scale accessibility is in fact harming local accessibility [18].

A more comprehensive review of active-accessibility measures is provided by [19]. Focusing on non-motorized accessibility, similar measures have been adopted, with some works using gravity models [20] and others adopting distance based models [21]. Within this research area, many studies have highlighted the importance of incorporating different travel behaviors and abilities in accessibility measures, particularly in urban settings where demand is heterogeneous and diversified [22]. The accessibility stream discussed so far primarily focuses on developing metrics and testing them in descriptive studies. When the objective is to design and improve infrastructure, facility location (FL) and network design (ND) models become more appropriate. In this context, the most popular approach for measuring accessibility is to minimize total travel distance or to maximize demand coverage. The use of gravity models is less popular within Operations Research studies for several reasons. First, gravity-based models are typically best suited for decentralized settings where customers choose facilities not necessarily by proximity. In contrast, when there is a central authority aiming to minimize travel distances (such as in public school and health service assignments) a proximity-based approach is more suitable [23]. Second, gravity models can be computationally challenging as they typically feature non-linear impedance functions which are more challenging to incorporate into network

design problems. Finally, distance-based approaches are simpler to communicate with stakeholders. For instance, using distance or temporal thresholds to evaluate service coverage replicates popular policies, such as the concept of 15-minute cities, which are being pursued in Paris [24] and Bangkok [25].

The Facility Location (FL) problem aims to identify the optimal location of assets within a system to optimize a given performance criterion (e.g., minimizing construction and operating costs). Past research studies on accessibility have proposed a variety of models extending p-median and p-center problems [26] and the fixed-charge facility location problem [27]. For example, [28] propose a model to maximize accessibility to services by identifying both the locations and capacities of facilities, incorporating user-equilibrium conditions to account for congestion. Similarly, [29] introduce a methodology to compute spatial accessibility and integrate it with location models to improve access to health facilities. A multi-objective problem to find healthcare facility locations is studied by [30]. They aim to maximize gravity-based accessibility function while simultaneously reducing other factors such as, inequality of access, number of people outside service radius and the building costs. [31] formulate a healthcare facility location problem accounting for demand coverage, equity and accessibility. [32] propose a FL model to maximize accessibility measured as a combination of different indicators such as service coverage, number of opportunities, travel cost, distance and spatial disaggregation. Facility location and walkability issues in the context of floodings are discussed by [33]. They propose a methodology to locate shelters accessible to people by walking. Their approach focus on optimizing criteria such as accessibility, safety and shelters' capacity. [34] introduce a facility location problem to add multi-modal mobility hubs to increase intermodal accessibility in rural areas.

The Network Design (ND) problem focuses on identifying which links to build or upgrade to optimize network performances. For instance, [35] propose a formulation to identify the road segments to upgrade to withstand all weather types thereby increasing service coverage. A similar arc-upgrading problem is studied by [36], with the objective to increase the capacity of arcs so that larger flows can be transported over shorter paths. The concept of equity is incorporated into an arc-upgrading problem by [37]. They aim to maximize different metrics that measure accessibility-based equity and minimize capacity construction costs on a User-Equilibrium network. Focusing on space and time accessibility on a transportation system, [38] propose a model to choose which links to construct so as to minimize the number of travellers that cannot reach a location and perform an activity within a travel budget. The concept of travel budget is also featured in the model introduced by [39]. They focus on a road network using user equilibrium principles and consider both deterministic and stochastic demand. Bike-specific design issues such as reducing intersections and increasing level of service are incorporated into a network design problem by [40] alongside with a standard accessibility metric based on travel time budget. [41] focus on a generic transportation network with the objective of minimizing the amount of inaccessible origin-destination pairs.

A third stream combines FL and ND, referred to as the Facility Location Network Design Problem (FLND). While many studies investigate the FLND in its generic form [42, 43], few specifically address accessibility issues. Among them [44, 45] introduce FLND formulations to optimize costs, accessibility, and equity by locating healthcare facilities and designing transport networks.

Another crucial aspect is the presence of different types of services and facilities. Multi-type facility location and network design problems have received substantial interest [46, 47], but few works focus on accessibility. [48] develop a location model to reduce the load of hospitals by locating two types of primary healthcare centers. [44] incorporate several healthcare services in their FLND model.

While several studies model different travel times in network design problems [49, 50], they often focus on multimodal transportation and travel choice behavior rather than on diverse user needs and abilities. The concept of accessibility for different demand groups has been mostly studied from a descriptive point of view [51].

In summary, while the majority of location and network design optimization models have focused on variants of distance-based accessibility, this is likely due to the centralized planning perspective often adopted in such studies. Although the body of FLND literature is extensive, it lacks a specific focus on accessibility, particularly for non-motorized transportation. We propose a model that expand current literature by integrating heterogeneous demand data to account for a range of travel abilities and facilities offering different types of services. The problem is solved using a Greedy Randomized Algorithm. An application to Huai Khwang district in Bangkok is presented to highlight the potential benefits of using the model as a decision-support tool.

## Optimization model to improve accessibility

In this section, we formulate the problem of jointly locating new facilities and expanding the non-motorized accessibility network as an integer optimization model. The network is represented as a graph $G(N, E)$, where $N$ is the set of nodes (i.e., facilities and demand points) and $E$ is the set of edges. A budget $l$ is available to improve the network by adding new accessible edges, grouped into the set $E'$. Binary variable $z_e$ is used to model the decision of adding edge $e$ to the network. Nodes can be demand points, indexed by $s \in S \subset N$, and destination facilities, indexed by $d \in D \subset N$. Each demand point belongs to a demand segment $p \in P$, which define the ability to travel along any edge and the travel time. The primary objective of this work is to enhance non-motorized accessibility rather than address potential congestion issues, which is why we assume uncapacitated facilities. Formally, an edge $e$ requires $t_{ep}$ units of time to be traveled by users in group $p$. We do not assume uncertainty in travel time but rather focus on incorporating distinct travel speeds with each demand groups. Adding new edges to the network does not necessarily means building entirely new connections. It also models the process of upgrading existing edges so that they are made accessible to a new group of users. For example, a sidewalk may not be accessible to some groups if its slope is overly steep. Subset $E'_p$ contains all the edges that can be added to the network from the point of view of demand group $p$. Set $E'$ groups all the edges that are non-accessible from at least one demand group. We further assume that a total of $n$ additional facilities can be chosen from a set of candidate locations $d \in D'$. Different types of facilities offering various services exist, and sets $D(k)$ group all the existing facilities providing type $k$ service. Binary variables $w_{dk}$ indicate whether a $k$-type facility is built at location $d$.

A node $s$ is covered by service $k$ if there is at least one working path connecting the node to a facility of type $k$, within a given time threshold $\tau$. We refer to a path as *working* if it already exists or all the new edges part of it are added to the network. The overall list of parameters, sets, indices, and variables is provided below.

**Set and indices**

- $E$ is the set of existing edges, indexed by $e$

- $E'$ is the set of candidate edges, indexed by $e$

- $E'_p$ is the set of candidate edges according to demand group $p$, indexed by $e$

- $N$ is the set of nodes of the network, indexed by $s$ and $d$

- $S \subset N$ is the set of demand nodes, where trips are originated, indexed by $s$

- $K$ is the set of types of facilities, indexed by $k$

- $D(k) \subset N$ is the set of existing facilities of type $k$, indexed by $d$

- $D' \subset N$ is the set of candidate locations for new facilities

- $P$ is the set of demand groups, indexed by $p$

**Parameters**

- $l_e$ is the length of edge $e$

- $t_{ep}$ is the traveling time of edge $e$ for demand in group $p$

- $l$ is the length budget available to add new edges to the network

- $n$ is the maximum number of facilities that can be added to the network

- $a_{sk}$ is the demand for service of type $k$ at location $s$

- $\tau$ is the maximum time considered to access a facility

- $p_s$ is the demand group of $s$

**Decision variables**

- $x_{sde}$ is equal to 1 if edge $e$ is on a path connecting $s$ to $d$; 0 otherwise

- $y_{sdk}$ is equal to 1 if service of type $k$ is provided to $s$ by facility $d$; 0 otherwise

- $z_e$ is equal to 1 if $e \in E'$ is added to the network; 0 otherwise

- $w_{dk}$ is equal to 1 if a facility of type $k$ is placed at candidate location $d$

A mixed-integer formulation of the Facility Location Network Design problem to improve Accessibility (FLNDA) is given below:

$$[\text{FLNDA}] \max_{\mathbf{x,y,z}} \sum_{s \in S} \sum_{k \in K} \sum_{d \in D(k) \cup D'} a_{sk} y_{sdk} \tag{1}$$

$$\text{s.t.} \quad \sum_{d \in D(k) \cup D'} y_{sdk} \leq 1 \quad \forall s \in S, k \in K \tag{2}$$

$$\sum_{e \in E'} l_e z_e \leq l \tag{3}$$

$$\sum_{k \in K} \sum_{d \in D'} w_{dk} \leq n \tag{4}$$

$$\sum_{k \in K} w_{dk} \leq 1 \quad \forall d \in D' \tag{5}$$

$$\sum_{e \in FS(i)} x_{sde} - \sum_{e \in BS(i)} x_{sde} = \begin{cases} 1 & \text{if } i = s \cap d \in D' \\ -1 & \text{if } i = d \cap d \in D' \\ \sum_{k \in K} w_{dk} & \text{if } i = s \cap d \in D \\ -\sum_{k \in K} w_{dk} & \text{if } i = d \cap d \in D \\ 0 & \text{if } i \neq s, d \end{cases} \tag{6}$$

$$x_{sde} \leq z_e \quad \forall s \in S, d \in D \cup D', e \in E'_{p_s} \tag{7}$$

$$y_{sdk} \leq w_{dk} \quad \forall s \in S, k \in K, d \in D' \tag{8}$$

$$y_{sdk} \leq 1 + \frac{\tau - \sum_{e \in E \cup E'} t_{ep_s} x_{sde}}{M} \quad \forall s \in S, k \in K, d \in D(k) \cup D' \tag{9}$$

$$x_{sde} \in \{0, 1\} \quad \forall s \in S, d \in D \cup D', e \in E \cup E' \tag{10}$$

$$y_{sdk} \in \{0, 1\} \quad \forall s \in S, k \in K, d \in D(k) \cup D' \tag{11}$$

$$z_e \in \{0, 1\} \quad \forall e \in E' \tag{12}$$

$$w_{dk} \in \{0, 1\} \quad \forall d \in D', \forall k \in K \tag{13}$$

The objective of the model is to maximize the amount of covered demand (Eq 1), accounting for different types of services. Eq (2) ensure that demand is not counted multiple times when $s$ is covered by more than one facility for the same service. Eq (3) limits the overall length of new edges that can be added to the network. Eq (4) limits the maximum number of new facilities. Eq (5) prevents multiple services from being co-located. Eq (6) are classic balance flow constraints with two additional equations needed to account for cases where destinations are candidate facilities. Specifically, if facility $d$ is not built (i.e., $\Sigma_{k \in K} w_{dk} = 0$) all the $x_{sde}$ variables are set to 0 and no path is built. If facility $d$ is built (i.e., $\Sigma_{k \in K} w_{dk} = 1$), the right hand side of the equation concides to the standard flow balance values.

Eq (7) prevents a non-existing edge to be used by $s$ unless it is added to the network. Similarly, Eq (7) state that $s$ can use $d$ for service $k$ only if a facility for service $k$ exists in that location. Eq (9) guarantee that $s$ is covered for service $k$ at location $d$ only if a working path shorter than $\tau$ exists between $s$ and $d$. Finally, constraints (10)–(13) specify the binary requirements for the decision variables.

## Solution methodology

Solving FLNDA problem is challenging as Facility Location and Network Design problems are known to be NP-hard [42]. Consequently, general purpose solution algorithms are only suitable to tackle small networks. For this reason, a randomized greedy heuristic is developed to provide near optimal solutions for instances of larger sizes. The choice of a randomized greedy algorithm is based on two key factors. First, empirical assessments of optimal solutions have shown that they are often close to those generated by greedy algorithms, making a greedy-inspired approach a logical starting point. Second, the simplicity of the randomized greedy algorithm enhances its ease of communication to relevant stakeholders and facilitates straightforward implementation. A flowchart of the main steps of the algorithm is depicted in Fig 1.

The first step of the Randomized Greedy procedure is to populate the set of new facilities $\hat{D}$ by creating a list of promizing locations with the *Rank_Facilities* procedure. The list is made of the best location and service pairs, assessed in terms of added demand coverage. One pair is randomly selected and added to $\hat{D}$ and the process is repeated until $n$ facilities are added. With $\hat{D}$ given, $it_2$ iterations are run to discover the best set of candidate edges to add to the network. Similarly to the first step, this process requires the creation of a second list using procedure *Rank_Edges*. New candidate edges are randomly selected from the list and added to $\hat{E}$ until the budge $l$ is fully utilized. Having pre-computed paths and given the sets of facilities $\hat{D}$ and edges

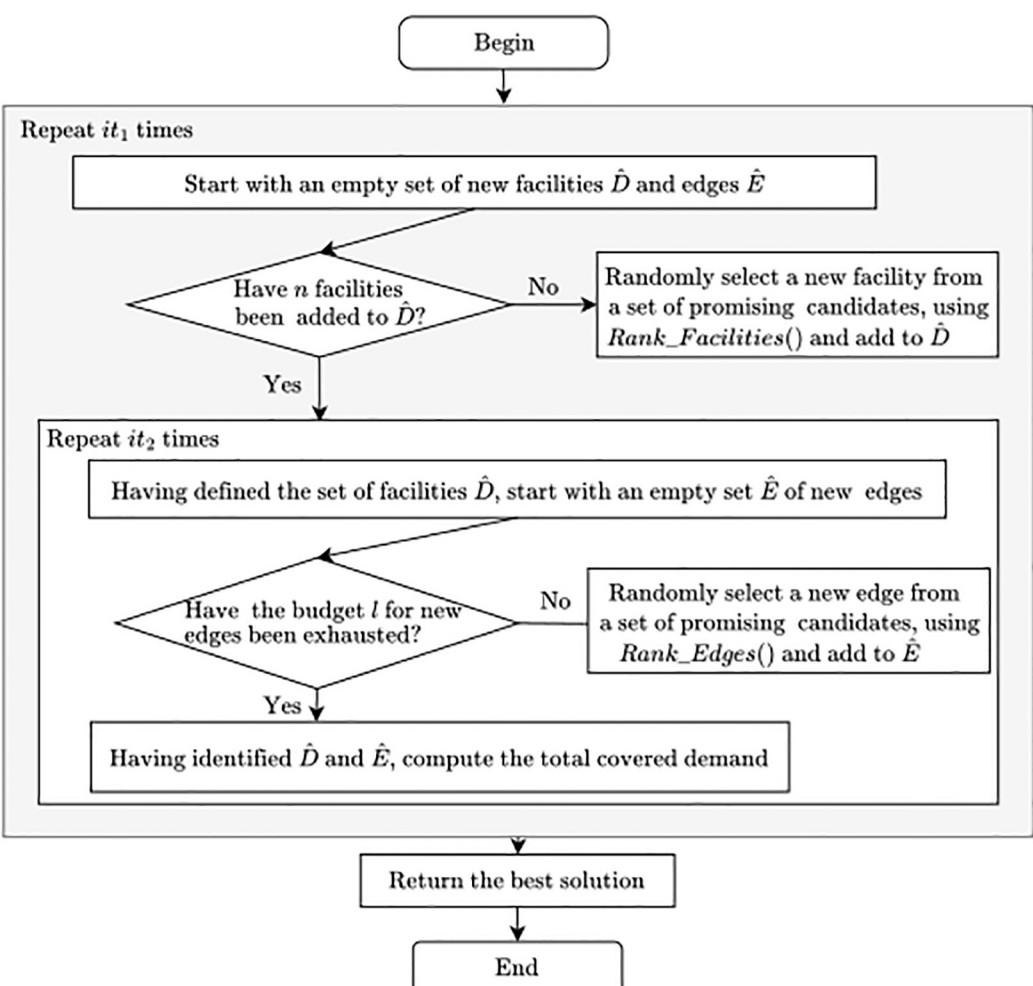

**Fig 1. Flowchart diagram for the randomized greedy heuristic.**

$\hat{E}$, the covered demand $obj$ can be directly computed and the best objective $obj^*$ and solution $sol^*$ can be updated. The overall process is repeated $it_1$ times, and the best solution is returned.

**Algorithm 1: Rank_Facilities**

```
Data: D̂, ρᴰ
for d ∈ D′ do
  if d ∉ D̂ then
    Ê ← {}, l̂ ← l;
    while l > 0 do
      Randomly select ê from e ∈ E(d) ∩ E′ \ Ê and add to Ê;
      l ← l − l_ê;
    for k ∈ K do
      Add (d, k) to D̂;
      Compute and store objective function obj given D̂ and Ê;
      Remove (d, k) from D̂;
Create a list by selecting the top-ρᴰ (d, k) pairs based on obj values;
return top-ρᴰ (d, k) pairs;
```

The *Rank_Facilities* procedure compute the objective value of each $(d, k)$ pair added to the input set $\hat{D}$. However, given that this procedure is run before any new edge is selected, a

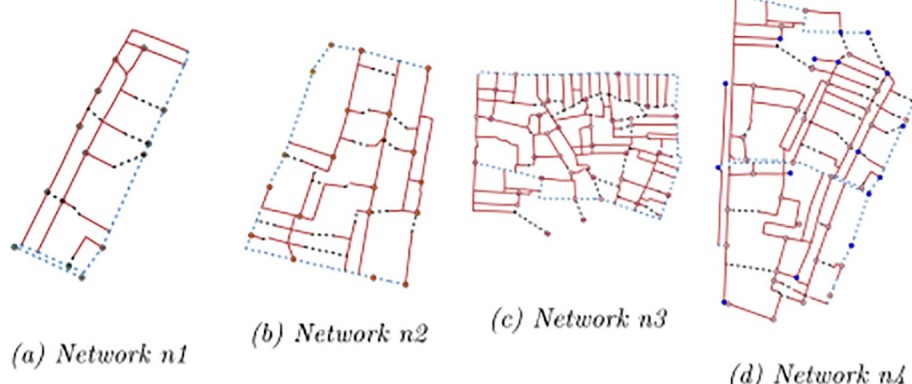

(a) Network n1  (b) Network n2  (c) Network n3  (d) Network n4

**Fig 2. Test networks for computational analyses.**

number of demand points that are partially or entirely disconnected from the existing network could be potentially disregarded. To address this issue, every time a new location $d$ is assessed the algorithm firstly randomly add as many new edges incidents to node $d$, grouped by set $E(d)$, as the budget $l$ allows. Then the objective is computed and the top-$\rho^D$ scoring pairs are added to the list.

**Algorithm 2: Rank_Edges**

```
Data: D̂, Ê, ρᴱ
for e ∈ E′ ∪ Ê do
  for k ∈ K do
    Add e to Ê;
    Compute and store objective function obj given D̂ and Ê;
    Remove e from Ê;
Create a list by selecting the top-ρᴱ edges based on obj values;
return top-ρᴱ edges;
```

Finally, the *Rank_Edges* procedure simply ranks any candidate edge measuring the increase in demand coverage when such edge is added to set $\hat{E}$. A list of the top-$\rho^E$ edges is returned.

The solution approach is tested on four networks (Fig 2) of different sizes. These instances are obtained as portions of a larger network used as a case study and detailed in Section 4.

The main features of the test networks are summarized in Table 1. All four instances assume 3 types of customers and facilities offering 2 services. The main differences are in the size of the problems, with the network having different values of candidate edges, facility locations and demand points.

We conduct a comparative analysis between Gurobi solver and the Randomized Greedy algorithm. Both approaches are implemented in Python using a PC with AMD Ryzen 7 3700X processor and 64GB of RAM with a 7,200 seconds runtime limit. The computational scenarios are built letting $n$ range between 1 and 5 and $l$ between 100 and 500. The parameters of the

**Table 1. Details of the networks used in the computational analysis.**

| Net | $|E|$ | $|E'|$ | $|S|$ | $|D'|$ | $|P|$ | $|K|$ |
|-----|-------|--------|-------|--------|-------|-------|
| n1 | 49 | 24 | 24 | 6 | 3 | 2 |
| n2 | 87 | 34 | 42 | 6 | 3 | 2 |
| n3 | 224 | 87 | 66 | 13 | 3 | 2 |
| n4 | 194 | 73 | 102 | 17 | 3 | 2 |

**Table 2. Randomized greedy parameters.**

|  | $n$ | | | | |
|---|---|---|---|---|---|
|  | **1** | **2** | **3** | **4** | **5** |
| $\rho^N$ | 3 | 4 | 6 | 6 | 6 |
| $it_1$ | 10 | 15 | 20 | 20 | 20 |

heuristic have been empirically set to the values provided in Table 2. Additionally, $\rho^E$ and $it_2$ are set to 6 and 15, respectively, across all scenarios.

Results are reported in Table 3. Columns $g$, show the relative gap between the solver and the heuristic objectives. Columns $t^S$ and $t^H$ are the computing time of the solver and heuristic, respectively. The star sign is used to highlight scenarios where the solver reached the runtime limit.

The table clearly highlights that commercial solvers like Gurobi are only suitable to tackle simple problems. Once the size of the network and budget parameters increase, the solver fails to converge, and the best found solution is used to calculate $g$. The heuristic perform well across the scenarios, retrieving optimal and near-optimal solutions. On average, $g$ is about 0.5%, with the highest value being 3.1%. In two occasions, the heuristic outperforms the solver, with $g$ taking negative values. The solver is generally more computationally expensive consistently demanding more time and resources, with the exception of the smallest networks $n1$ and $n2$. On the other hand, the heuristic is, on average, over 80% faster, with this advantage becoming particularly evident in complex instances.

**Table 3. Results of the computational analysis.**

|  |  | $l$ | | | | | | | | | | | | | | |
|---|---|---|---|---|---|---|---|---|---|---|---|---|---|---|---|---|
|  |  | **100** | | | **200** | | | **300** | | | **400** | | | **500** | | |
| **Net** | $n$ | $g$ | $t^S$ | $t^H$ | $g$ | $t^S$ | $t^H$ | $g$ | $t^S$ | $t^H$ | $g$ | $t^S$ | $t^H$ | $g$ | $t^S$ | $t^H$ |
| n1 | 1 | 0.0 | 3 | 19 | 0.0 | 7 | 24 | 0.0 | 28 | 42 | -0.0 | 40 | 55 | 0.0 | 38 | 63 |
|  | 2 | 0.0 | 4 | 24 | 0.0 | 8 | 27 | 0.3 | 28 | 55 | 0.3 | 33 | 62 | 0.0 | 32 | 72 |
|  | 3 | 0.0 | 3 | 25 | 0.0 | 5 | 31 | 0.3 | 8 | 63 | 0.3 | 2 | 70 | 0.2 | 3 | 74 |
|  | 4 | 0.0 | 3 | 32 | 0.0 | 5 | 37 | 0.0 | 8 | 72 | 0.2 | 3 | 82 | 0.0 | 3 | 92 |
|  | 5 | 0.2 | 2 | 35 | 0.1 | 6 | 42 | 0.0 | 10 | 83 | 0.0 | 3 | 81 | 0.0 | 3 | 93 |
| n2 | 1 | 0.3 | 20 | 17 | 0.0 | 47 | 51 | 0.1 | 91 | 64 | 0.6 | 229 | 83 | 0.0 | 156 | 101 |
|  | 2 | 0.0 | 16 | 25 | 0.0 | 43 | 70 | 0.4 | 156 | 109 | 0.2 | 270 | 130 | 0.2 | 387 | 149 |
|  | 3 | 0.0 | 16 | 28 | 0.0 | 30 | 97 | 0.2 | 134 | 131 | 0.4 | 475 | 145 | 0.2 | 379 | 168 |
|  | 4 | 0.0 | 18 | 29 | 0.1 | 33 | 94 | 0.5 | 111 | 158 | 0.3 | 255 | 176 | 0.4 | 396 | 213 |
|  | 5 | 0.0 | 17 | 38 | 0.2 | 30 | 108 | 0.7 | 105 | 181 | 0.6 | 85 | 192 | 0.2 | 120 | 211 |
| n3 | 1 | 0.0 | 501 | 176 | 0.0 | 1,428 | 551 | 0.9 | * | 850 | 0.6 | * | 894 | 1.1 | * | 1,456 |
|  | 2 | 0.0 | 209 | 245 | 0.7 | 696 | 511 | 0.5 | * | 989 | 0.7 | * | 1,047 | 0.3 | * | 1,110 |
|  | 3 | 0.5 | 206 | 300 | 0.0 | 813 | 756 | 0.2 | * | 1,212 | 0.2 | * | 1,366 | 0.5 | * | 1,405 |
|  | 4 | 0.2 | 238 | 451 | 0.4 | 2,417 | 832 | 2.1 | * | 1,315 | -0.7 | * | 1,505 | 1.3 | * | 1,772 |
|  | 5 | 3.3 | 182 | 245 | 3.1 | 2,486 | 896 | 2.0 | * | 1,457 | 0.8 | * | 1,456 | 1.0 | * | 1,727 |
| n4 | 1 | 0.0 | 3,072 | 81 | 0.0 | * | 214 | 0.0 | * | 390 | 0.5 | * | 395 | 0.7 | * | 383 |
|  | 2 | 0.2 | 1,862 | 138 | 1.0 | * | 358 | 0.1 | * | 644 | 0.6 | * | 604 | -0.6 | * | 653 |
|  | 3 | 0.7 | 1,001 | 198 | 0.1 | 6,445 | 509 | 0.4 | * | 967 | 2.3 | * | 791 | 0.6 | * | 819 |
|  | 4 | 1.1 | 726 | 248 | 0.9 | 4,799 | 680 | 1.1 | * | 1,098 | 2.6 | * | 917 | 1.9 | * | 984 |
|  | 5 | 0.5 | 411 | 297 | 0.2 | 2,772 | 761 | 1.6 | * | 1,196 | 3.0 | * | 1,092 | 2.0 | * | 1,120 |
|  |  | **0.4** | **426** | **133** | **0.3** | **1,815** | **332** | **0.6** | **3,642** | **554** | **0.7** | **3,678** | **557** | **0.5** | **3,684** | **633** |

## Improving accessibility in Huai Kwang district

In this section, a case study based on Huai Khwang district in Bangkok is discussed. Focusing on Bangkok is particularly interesting because the city has experienced rapid urbanization and population growth, which have significantly strained its transportation systems and infrastructure. Bangkok has a high rate of car ownership and a culture that heavily relies on personal vehicles, which has led to traffic congestion and air pollution. Additionally, Thailand is among the fastest ageing countries in the world [52] There is an urgent need for sustainable long-term transportation solutions to improve people's liveability. Currently, Bangkok has embarked on a series of projects to expand pavements and building accessible ramps [53], showing the significance of non-motorized accessibility in the city. Huai Khwang is among the most populated districts in Bangkok with a population of about 80,000. This district has a combination of high rise residential and office projects as well as a number of congested residential areas. Pedestrian infrastructure in this district poses significant challenges for non-motorized mobility, particularly for individuals with mobility impairments. Sidewalks, where they exist, are often narrow, uneven, and cluttered with obstacles such as light poles, concrete barriers, and abrupt steps, complicating pedestrian navigation. Moreover, walking can be dangerous at crossings if no pedestrian lights are available or enforced. An accessibility network of the district is developed with QGIS using a combination of data sources. The network is built with OpenStreetMap data and every edge is then inspected with Google Street View and labelled as accessible, pedestrian-only and non-walkable. An edge is deemed accessible if it can be safely traveled by both pedestrians and individuals with limited mobility. Edges are classified as pedestrian-only when obstacles prevent access by those with mobility impairments, yet remain navigable for pedestrians. Conversely, an edge is marked as non-walkable if it poses safety risks to all users, such as sections with high-speed traffic without sidewalks or intersections lacking traffic signals. Additionally, we empirically identified and added several candidate edges to the network to explore options for enhancing connectivity. Demand points are constructed using buildings data from [54]. Over 9000 buildings are clustered using $k$-means (with $k$ set to 100) and the centroid of each cluster is selected as a demand point. For each demand point $s$, the demand $a_{sk}$ is equal to the number of buildings within its cluster. We focus on two types of services: healthcare and schools. Existing facilities in these categories are integrated into the network. Candidate locations for both services are placed where clear plots of undeveloped land are available. Demand is categorized into three groups: healthy pedestrians (H), elderly pedestrians (E) and people with mobility impairments (MI). The latter group is specifically unable to navigate edges classified as pedestrian-only. Each group has a different traveling speeds [55] and demand shares as defined in Table 4.

The resulting network, depicted in Fig 3a, has 300 demand points, 5 schools and 4 healthcare centers, 40 candidate locations, 732 edges of which 139 are non-walkable or non-existing. The network portrayed on the right in Fig 3b exclusively displays accessible links, showing its fragmented nature, particularly from the point of view of mobility-impaired users who face significant limitations in reaching many of the available destinations.

The first analysis aims to measure the impact on total demand coverage of expanding the accessible network and introducing new facilities. The problem is solved for $n = 0 \ldots 5$ and

**Table 4. Traveling speed and demand share of different demand groups.**

| Healthy (H) | | Elederly (E) | | Mobility Impaired (MI) | |
|---|---|---|---|---|---|
| Speed (m/s) | Demand Share (%) | Speed (m/s) | Demand Share (%) | Speed (m/s) | Demand Share (%) |
| 1.2 | 75 | 0.8 | 20 | 0.5 | 5 |

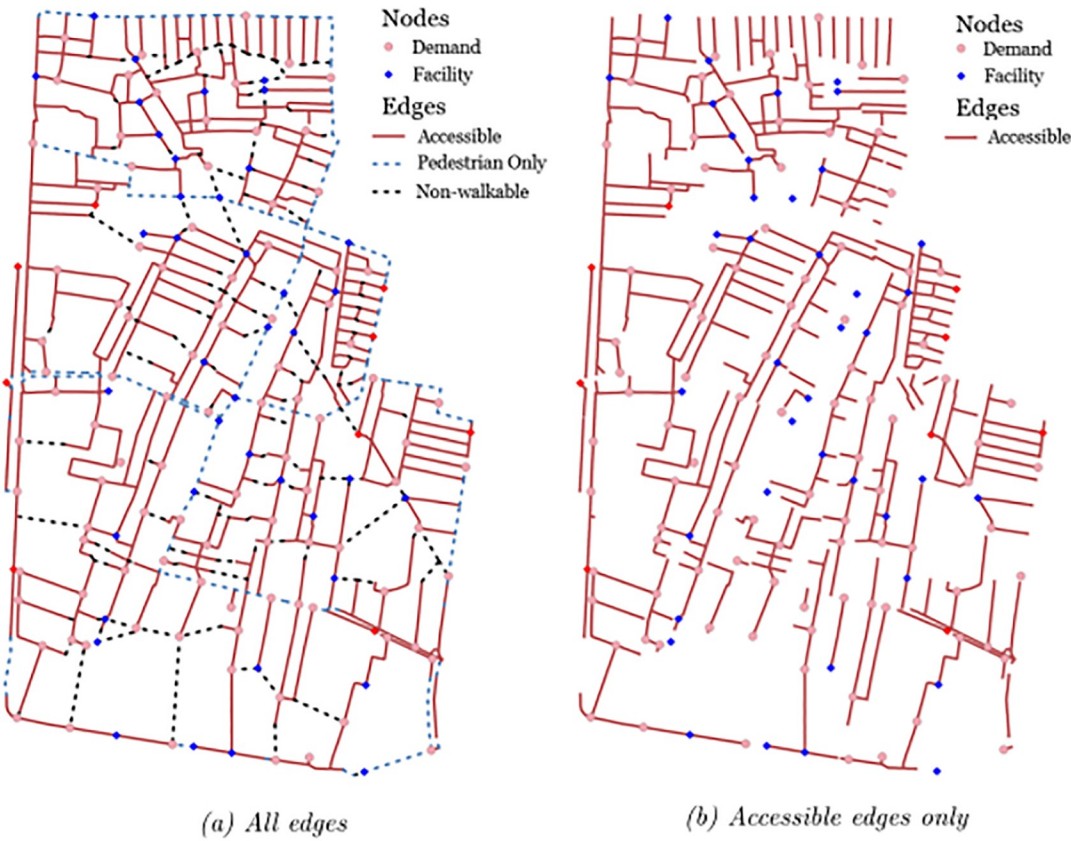

**Fig 3. Huai Kwang network.** (a) All edges (b) Accessible edges only.

$l$ = 0, 100 and 500 meters. Results are measured as the share of covered demand and reported in Fig 4. The graph reveals that existing facilities currently cover only 59% of the total demand. A modest investment of 500 meters of newly accessible links leads to a substantial increase in demand coverage, raising it to 73%. This is particularly significant for healthcare, which increase its coverage from 49% to 65%. However, more pronounced improvements require the introduction of new facilities. For instance, when 2 new facilities are added, both school and healthcare demand exceed 80% coverage, provided additional accessible links are available. Further investments continue to yield positive impacts, albeit with diminishing returns. Ultimately, achieving more than 90% coverage for both types of services needs to establish at least 5 new facilities.

Fig 5 illustrates accessibility changes, comparing the distribution of covered points by service type with no network and facility changes (i.e., $l$ = 0, $n$ = 0) with the scenario assuming the largest budget considered (i.e., $l$ = 1000, $n$ = 5). The red circle and blue square markers show the covered points for school and health services, respectively, for three different demand groups: H, E, and MI. The figure highlights how the expansion of the network and the addition of facilities lead to a better coverage across all groups, especially for health services in groups H and E.

A more detailed look on the incremental gains in demand coverage with the allocation of additional resources is provided by Fig 6. This analysis can support decision makers in assessing the best allocation of resources to the project. The chart depicts the percentage gains (and

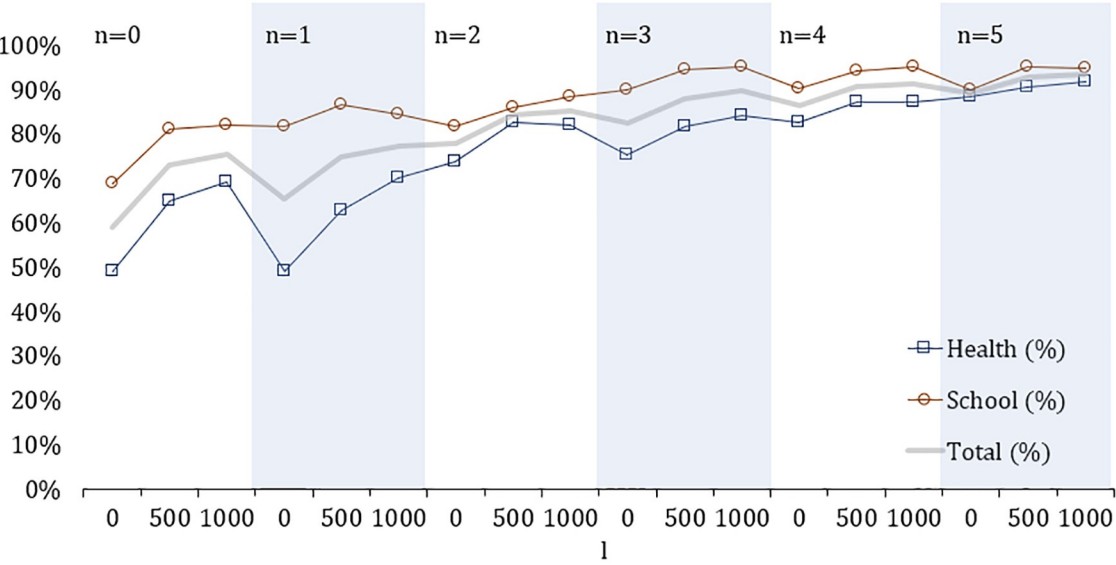

**Fig 4. Covered demand(%) for different values of _l_ and _n_.**

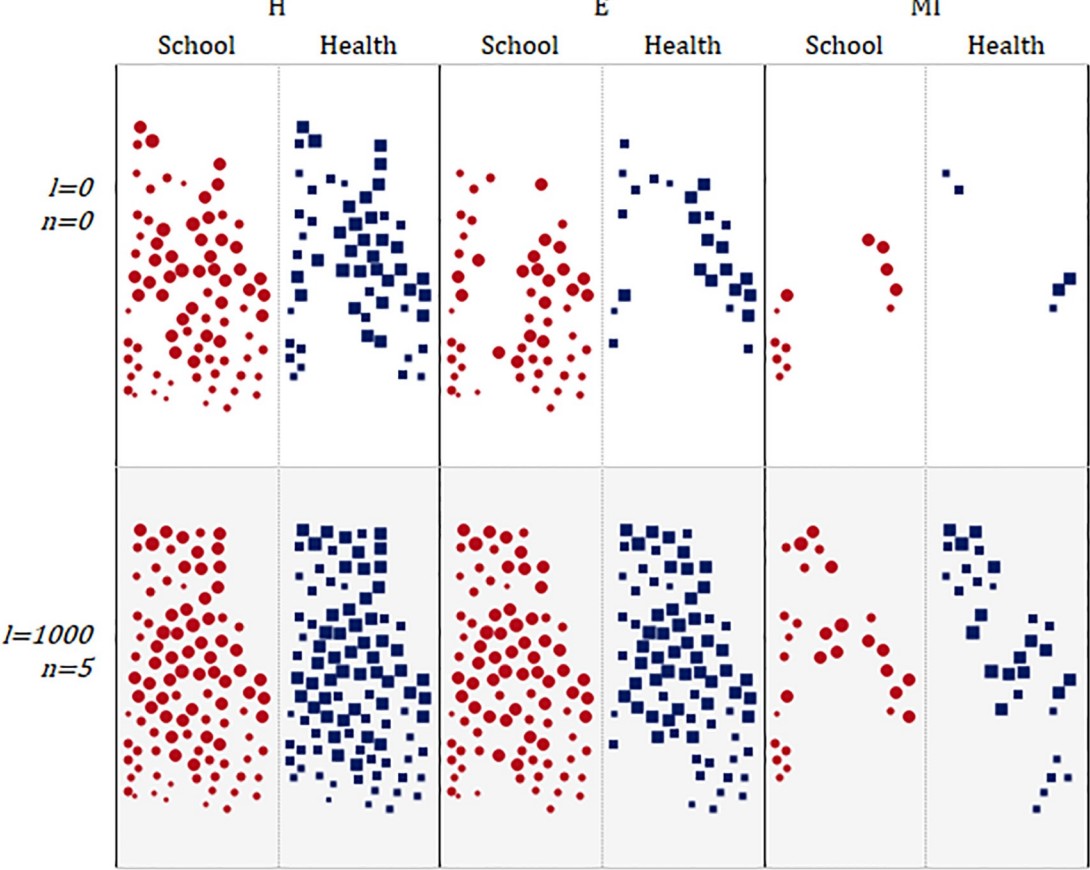

**Fig 5. Demand points for H, E and MI groups, covered by school and health services.**

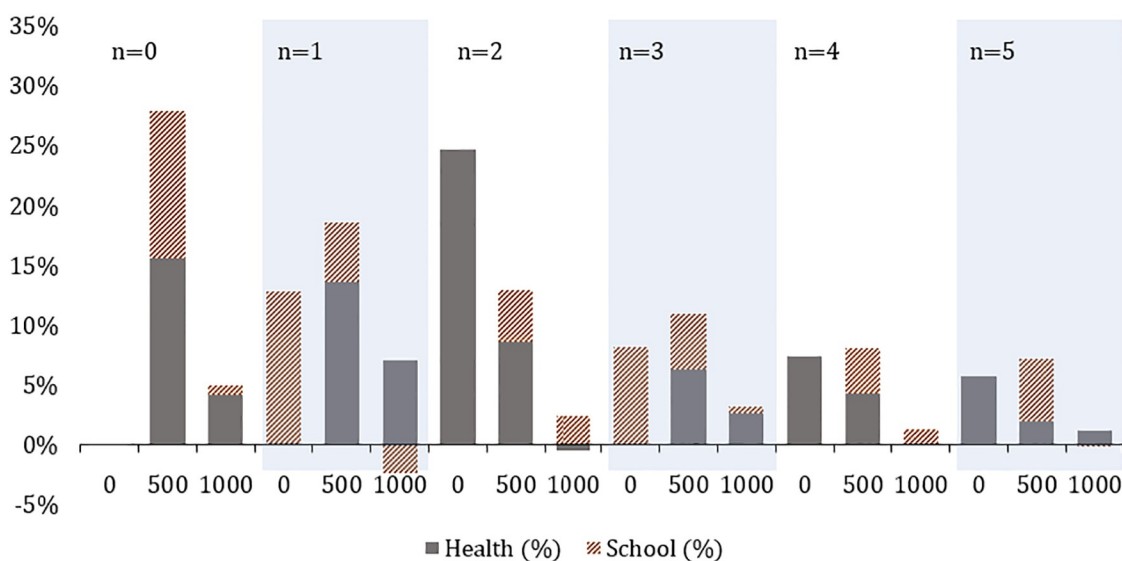

**Fig 6. Covered demand increments(%) for different values of *l* and *n*.**

losses) in covered demand obtained by increasing *l* and *n*. Specifically, bars corresponding to any *n*, *l* pair where *l* = 0, show the additional share of covered demand compared to the scenario when *n* − 1 and *l* = 0, thereby highlighting the value of introducing an additional facility. Other bars show the changes in demand coverage as more accessible links are added to the network. The chart shows that the impact of additional facilities and links diminishes as more resources are allocated. Moreover, increasing *l* from 500 to 1000 meters generally yields limited effects. Interestingly, modest investments in accessible links often yield greater gains in demand coverage than adding new facilities alone. For instance, the introduction of the first facility results in an additional 13% demand coverage for schools. When paired with 500 meters of accessible links, this coverage increases by an additional 5% and 14% for schools and healthcare services, respectively. Similar trends are observed for other values of *n*, with the exception of *n* = 2, where the benefit of adding a new health center surpasses other subsequent investments. This is due to the network having fewer health centers than schools. Overall, the results highlight the significant benefit of jointly designing network and facilities.

The following analysis aims to evaluate how individual demand groups are influenced by design decisions. More specifically, the objective of the analysis is measuring how designs made assuming a demand with a majority of healthy people, affect vulnerable groups such as elderly and mobility impaired individuals. We examine three distinct demand scenarios: the "ALL" scenario, where all demand groups (i.e., H, E, and MI) are considered; the "E" scenario, which assumes only elderly users; and the "MI" scenario, focusing solely on the mobility impaired group. The problem is solved for these scenarios, with *n* ranging from 1 to 5 while maintaining *l* = 500. Designs obtained with an *assumed* scenario are then tested against other *actual* demand scenarios to measure how distant they are from best known solutions. Results are reported as relative gaps in Table 5, showing the percentage drop in demand coverage. For example, column ALL/MI reports the gaps computed when designs obtained assuming ALL demand scenario are applied to MI actual demand scenario. Notably, considering ALL or E scenarios consistently leads to suboptimal design decisions from the perspective of MI users. For instance, the network design achieved with *n* = 4 and *l* = 500 results in 59% fewer MI users covered compared to the design obtained assuming only MI users. This disparity arises due to

**Table 5. Relative gaps between actual demand scenario Y and assumed scenario X (X / Y).**

| $n - l$ | ALL/E | ALL/MI | E/ALL | E/MI | MI/ALL | MI/E |
|---|---|---|---|---|---|---|
| 1-500 | 11% | 53% | 3% | 46% | 14% | 26% |
| 2-500 | 2% | 55% | 3% | 67% | 19% | 28% |
| 3-500 | 2% | 50% | 0% | 56% | 14% | 26% |
| 4-500 | 0% | 59% | 2% | 45% | 13% | 23% |
| 5-500 | 0% | 39% | 0% | 25% | 31% | 24% |

MI users representing a minority of the demand (i.e., 5%) and having distinct needs and requirements concerning network edges. On the other hand, there is some affinity between design plans for ALL and E scenarios motivated by the fact that the only difference between healthy and elderly users is in the walking speed.

Previous analysis highlighted potential issues in making decisions based on simple counts of demand, which may disproportionately impact smaller user groups, particularly those who are most vulnerable. Several techniques exist to address issues such as robustness and fairness; one straightforward approach available to decision-makers is to assign different weights to demand groups, thereby placing varying degrees of emphasis on each group. Formally, for a source $s$ belonging to group $p$, the demand $a_{sk}$ is multiplied by a scaling factor $\lambda_p$ set by the decision maker. The benefit of this approach is that it is simple and does not require changes in the formulation or the solution procedure. The objective of the analysis is to explore network designs that are less detrimental to MI users while maintaining a high overall demand coverage. To this aim, we set weights for H and E users equal to 1 and gradually increase the weight assigned to *MI* demand. We fix the budget at $n = 2$ and $l = 500$ and solve for different weight combinations. The results of the analysis are presented in Table 6. Similarly to Table 5, results are reported as relative gaps from the best solutions of the three demand scenarios under evaluations. Results show that there is potential for improvement, with progressively better solutions for MI users being discovered as more emphasis is placed on their share of demand. These improvements come with reasonably minor changes in the quality of network designs under other scenarios. For instance, when weights are set to (1, 1, 8), the demand coverage for ALL and E scenarios repsectively decreases by 2.8% and 13%; however, the gap in MI demand coverage from its best-known solution improves significantly from 55.1% to 16%.

In this analysis, we visualize network designs for different demand scenarios. Fig 7 depicts three networks showing the locations of new facilities and accessible edges selected by the best solutions for three demand scenarios: ALL, MI, and ALL with weights set to 1, 1, and 8. The first clear common pattern across scenarios is the location of new facilities in the central area of the district, addressing the evident gap left by existing facilities predominantly situated on the periphery. However, in the ALL scenario, the selected facilities are connected to lengthy pedestrian-only edges (dotted blue lines), while investments in edges are dispersed throughout

**Table 6. Relative gaps between solutions using different demand weights.**

| $\lambda_p$(H, E, MI) | ALL | E | MI |
|---|---|---|---|
| 1, 1, 1 | 0.0% | 2.3% | 55.1% |
| 1, 1, 2 | 1.3% | 1.9% | 45.4% |
| 1, 1, 4 | 1.0% | 3.9% | 33.2% |
| 1, 1, 6 | 2.1% | 10.4% | 25.4% |
| 1, 1, 8 | 2.8% | 13.0% | 16.0% |

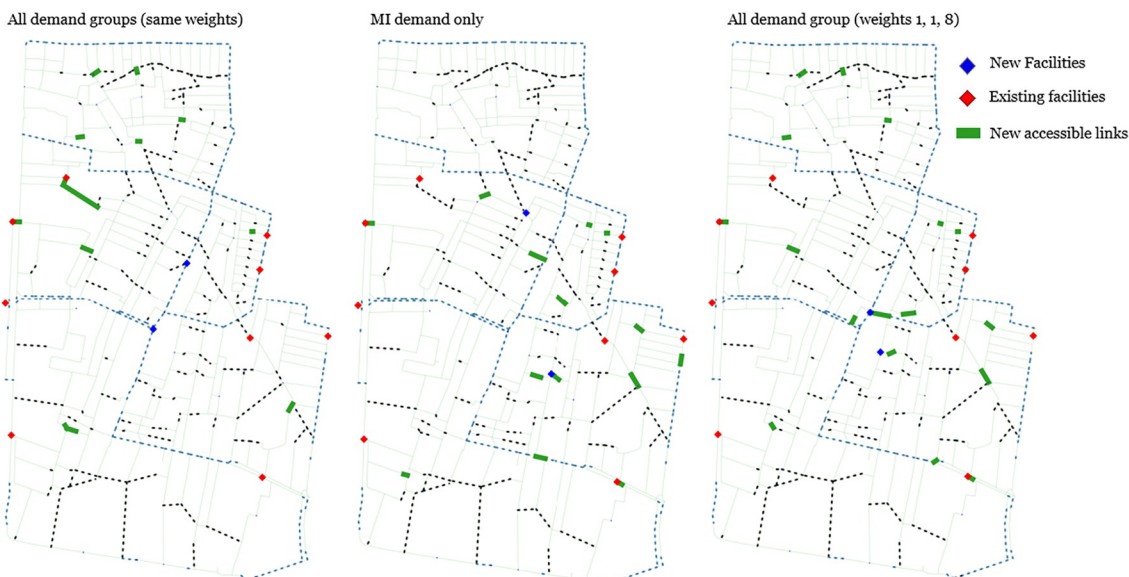

**Fig 7. Solution comparison between different demand group scenarios.**

the network, distant from the new facilities. Consequently, this design renders the newly added facilities inaccessible to MI users. Conversely, in the MI scenario, new facilities are either already linked to accessible paths or can be made accessible by adding short edges. Additionally, investments in edges are concentrated around the new facilities.

The weighted scenario represents a compromise between the two network plans. It entails opening two facilities in the central region, providing access to both while also investing into some peripheral edges that were selected by the ALL scenario.

Overall, it is evident that different design strategies may offer varying degrees of benefits for different user groups. The extent to which a decision maker is willing to prioritize robustness over other objectives ultimately depends on their specific goals and priorities. Finally, the complexity of this problem, even within the context of a relatively small-scale case study focused on a single district, supports the necessity of developing systematic tools to facilitate informed decision-making processes.

## Conclusions and future developments

In this work, we introduce an optimization model aimed at locating new accessible roads and service facilities, while accounting for demand groups with varying walking abilities. Recognizing the NP-hard nature of the model, computational analyses reveal that solving it using general-purpose solvers can be time-consuming, especially for instances of realistic sizes. To address this challenge, we develop and test a randomized greedy solution algorithm, demonstrating its efficacy in retrieving optimal and near-optimal solutions within reasonable time. The model is applied to a case study based on Huai Kwang district of Bangkok, focusing on health and education services. The results demonstrate that significant improvements can be achieved by adding new accessible roads along with new facilities. The case study also highlights differences in solutions for different demand groups and investigate a simple approach to obtain more robust designs. The model results help identify specific areas within the Huai Kwang district where adding new facilities and accessible roads would most improve accessibility for various demand groups. This shows that urban planners could use a similar approach

to guide zoning changes that promote the inclusion of essential services in underserved areas. Additionally, the findings support the need to enhance accessibility for vulnerable or mobility-impaired groups by prioritizing investments in accessible infrastructure development. The results align with the study's objective of developing a practical decision support tool for planners, demonstrating how targeted investments and strategic planning can significantly improve non-motorized accessibility.

While our work offers a novel approach to optimizing non-motorized accessibility, there are a few limitations to acknowledge. First, the model does not currently account for capacity constraints at facilities, which could impact accessibility in high-demand areas where congestion may occur. Second, although the heuristic developed is capable of handling realistic-sized networks, its scalability is challenged by significantly larger networks; addressing this may require developing more advanced algorithms or accepting a larger gap between optimal and retrieved solutions. Third, the model depends on detailed network data, requiring each link to be inspected and labeled with accessibility-relevant attributes. This data collection process can be demanding, especially in regions lacking granular data.

There are several lines of research that can be explored to expand this work. Firstly, the model could be expanded to incorporate a temporal dimension, enabling the construction of networks capable of accommodating significant demographic changes over time. Additionally, factors such as weather conditions and their impact on walkability could be explored, particularly in regions prone to issues like flooding. Integrating uncertainty and robustness features into the model could help address potential disruptions to paths caused by such environmental factors. Moreover, the problem scope could be broadened by considering a range of accessibility investments, such as building access ramps or adding cover-ways, to cater to different requirements for making links accessible. Finally, efforts should be directed towards refining solution algorithms to effectively tackle larger and more complex datasets as the problem complexity increases with the incorporation of new features.

## Author Contributions

**Conceptualization:** Stefano Starita.

**Data curation:** Stefano Starita.

**Formal analysis:** Stefano Starita.

**Funding acquisition:** Stefano Starita.

**Methodology:** Stefano Starita.

**Project administration:** Stefano Starita, Pinnaree Tea-Makorn, Pavitra Jindahra.

**Resources:** Pinnaree Tea-Makorn, Pavitra Jindahra.

**Software:** Stefano Starita.

**Validation:** Stefano Starita.

**Visualization:** Stefano Starita.

**Writing – original draft:** Stefano Starita, Pinnaree Tea-Makorn, Pavitra Jindahra.

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
