## [Decision Letter · Decision Letter 0]

6 Aug 2024

PONE-D-24-23480Building Non-Motorized Accessible Communities for Heterogeneous Demand: A Facility Location and Network Design ProblemPLOS ONE

Dear Dr. Starita,

Thank you for submitting your manuscript to PLOS ONE. After careful consideration, we feel that it has merit but does not fully meet PLOS ONE’s publication criteria as it currently stands. Therefore, we invite you to submit a revised version of the manuscript that addresses the points raised during the review process.

We look forward to receiving your revised manuscript.

Kind regards,

Abel C. H. Chen

Academic Editor

PLOS ONE

Journal Requirements:

3. Thank you for stating the following financial disclosure: "This research is supported by Rathchadapiseksompotch Fund Chulalongkorn University."

5. We note that Figure 2 in your submission contain [map/satellite] images which may be copyrighted. All PLOS content is published under the Creative Commons Attribution License (CC BY 4.0), which means that the manuscript, images, and Supporting Information files will be freely available online, and any third party is permitted to access, download, copy, distribute, and use these materials in any way, even commercially, with proper attribution. For these reasons, we cannot publish previously copyrighted maps or satellite images created using proprietary data, such as Google software (Google Maps, Street View, and Earth). For more information, see our copyright guidelines: http://journals.plos.org/plosone/s/licenses-and-copyright.

Reviewers' comments:

Reviewer's Responses to Questions

**Comments to the Author**

1. Is the manuscript technically sound, and do the data support the conclusions?

Reviewer #1: Yes

Reviewer #2: Partly

2. Has the statistical analysis been performed appropriately and rigorously? 

Reviewer #1: N/A

Reviewer #2: N/A

3. Have the authors made all data underlying the findings in their manuscript fully available?

Reviewer #1: No

Reviewer #2: No

4. Is the manuscript presented in an intelligible fashion and written in standard English?

Reviewer #1: Yes

Reviewer #2: Yes

5. Review Comments to the Author

Reviewer #1: The paper entitled Building Non-Motorized Accessible Communities for Heterogeneous Demand: A Facility Location and Network Design Problem is really interesting.

The needs to not only improving public transportation but also non-motorized infrastructure is indeed needed to some extends.

There is concern regarding the English quality on this manuscript as some parts are redundant, especially the background part. Furthermore, there are some parts that should be improved as i found some dangling modifier that reduce the overall paper clarity. Please check again and use English proofreading service to improve the quality and emphasizes the study aims and contributions.

I also notice that the tables & figures presented are somewhat unclear. Rather than using the notation l, n, in the figures or tables, I suggest the authors to use the actual variables. This will ensure better readability.

Please also make one more subsection focusing on managerial implication of the findings.

Reviewer #2: Please see the ATTACHED review report.

The paper is well-structured and addresses a significant and timely issue in urban planning. The combination of facility location theory and network design to enhance non-motorized accessibility is innovative and has the potential to contribute significantly to the field. However, the manuscript could benefit from a clearer exposition in some sections, more rigorous validation of the results, and deeper integration of the literature to position its contributions more distinctly within the current research landscape.

6. PLOS authors have the option to publish the peer review history of their article (what does this mean?). If published, this will include your full peer review and any attached files.

Reviewer #1: No

Reviewer #2: No

---

## [Author Response · Author response to Decision Letter 0]

18 Sep 2024

Referee Report Regarding the Manuscript: PONE-D-24-23480 Building Non-Motorized Accessible Communities for Heterogeneous Demand: A Facility Location and Network Design Problem Submitted to: PLOS ONE 

We sincerely thank the referees and editors for their valuable and constructive feedback. We have carefully considered your comments in revising the manuscript and believe that the revised draft has significantly improved as a result. Please find below our detailed responses (in red) to each of your comments.

A. Brief Summary This manuscript presents an optimization model to enhance non-motorized accessibility in urban environments by identifying optimal locations for service facilities and accessible road segments. The study uses a randomized greedy algorithm to address a complex network design and facility location problem, demonstrating its application through a case study in the Huai Kwang district of Bangkok. This research is timely due to increasing urban mobility issues and aims to cater to diverse user needs, enhancing access to essential services like healthcare and education. 

B. Overall Impression The paper is well-structured and addresses a significant and timely issue in urban planning. The combination of facility location theory and network design to enhance non-motorized accessibility is innovative and has the potential to contribute significantly to the field. However, the manuscript could benefit from a clearer exposition in some sections, more rigorous validation of the results, and deeper integration of the literature to position its contributions more distinctly within the current research landscape.

C. Major Comments 

1. Scientific Merits and Consistency of Arguments 

o The manuscript effectively frames the problem within the context of social justice and environmental sustainability. However, the linkage between the proposed model and existing theories could be strengthened. For instance, the discussion on non-motorized accessibility measures could be better integrated with the model’s objectives to highlight its innovation (e.g., how the proposed measures improve upon or differ from those like distance-based or gravity-based measures). 

We have revised the introduction section and background to better integrate our proposed model with existing theories on accessibility measures. The revised section clarifies the distinction between traditional distance-based and gravity-based measures and our optimization model, which is designed for centralized planning contexts. We highlight the limitations of gravity-based models in such settings due to their computational challenges and decentralized assumptions, emphasizing instead the practicality and policy alignment of distance-based approaches. Our model innovates by incorporating granular demand data to account for heterogeneous user groups, unlike many existing studies that focus on homogeneous demand or single-service types. This approach is particularly suited for urban planning scenarios that prioritize equity and accessibility, as seen in concepts like the "15-minute city." 

o The methodological explanation in Sections 2 and 3 is complex and could benefit from additional diagrams or flowcharts to aid in understanding the model’s workflow and algorithmic steps. 

We have replaced the pseudocode of the main algorithm with a higher level flow chart and improved the explanation of the procedures. We have retained the pseudocode of Rank_Facilities() and Rank_Edges() as we believe they are short and simple enough to be digested by a non-technical audience. 

o In regards to the optimization model, the assumptions and limitations should be explicitly stated. For instance, how does the model handle uncertainties in demand and travel time? What are the implications of these assumptions on the model’s applicability in real-world scenarios? 

We have added a paragraph on the limitations of our study in the conclusions, discussing key assumptions and limitations of the model, such as the absence of capacity constraints at facilities, the challenges associated with very large networks, and the demanding data collection process. We have also enhanced the discussion of our assumptions while introducing the model and provided a rationale for not considering uncertainty in travel times, stating: “We do not assume uncertainty in travel time because our approach incorporates varying travel times through distinct demand groups, each reflecting different user behaviors and speeds.” Additionally, we have expanded our discussion on future directions, highlighting that uncertainty in demand, particularly with long-term demographic changes, could significantly impact the problem and offers an interesting direction for further research.

o The choice of a randomized greedy algorithm needs more justification. Why was this specific algorithm chosen over others? Comparative analysis with other potential algorithms would strengthen the argument for its use

We have clarified the rationale for choosing a randomized greedy algorithm in the revised Solution Methodology section. The decision was guided by empirical observations that optimal solutions are often close to those derived from greedy algorithms, making a greedy-inspired heuristic an effective starting point. Additionally, the algorithm's simplicity ensures it is easier to communicate to stakeholders and implement in practice. While we agree that a comparative analysis with other potential algorithms could provide additional insights, we believe that such an analysis would make the paper overly long and too technical for the intended audience of this journal. However, this could be a valuable direction for future research.

2. Contributions to the Field 

o The paper's contributions to urban planning and accessibility could be articulated more compellingly by comparing the results with other contemporary approaches. This could include a discussion on the scalability of the approach to other countries, which is currently understated. Please highlight the novel aspects of your work more clearly in relation to existing studies. 

We have revised the manuscript to better highlight the novel aspects of our work and address the scalability of our approach.

Novelty of the Model: Our model presents a novel Facility Location and Network Design (FLND) formulation that incorporates different travel groups and speeds, which has not been explored in the accessibility literature. This approach addresses gaps identified in existing studies, offering a more inclusive and realistic tool for urban planning. We have emphasized this contribution more clearly in the revised manuscript, particularly in the literature review section.

Scalability: Our approach is adaptable to various contexts as it does not depend on any specific country or policy framework. The methodology and data collection processes can be replicated in different regions, making the model applicable globally to address different urban accessibility challenges.

3. Quality and Consistency of Writing, Article Flow, and Presentation of Results 

o Some sections, especially the methodology and results sections, suffer from dense presentation. Simplifying these descriptions or using supplementary material to shift detailed algorithmic discussions could enhance readability. 

As mentioned in our previous response regarding the complexity in Sections 2 and 3, we have replaced the pseudocode of the main algorithm with a higher-level flow chart. Additionally, we have enhanced the discussion of the algorithm to ensure it is more accessible and understandable to a non-technical audience.

o The presentation of results (Section 4) could be enhanced by more visual representations of the changes in accessibility patterns before and after the implementation of the model. 

We have enhanced the presentation of the results in the case study Section by adding a visual representation that illustrates the changes in accessibility patterns before and after implementing the model. Specifically, we included a figure that compares the distribution of covered points by service type with no changes to the network and facilities versus the scenario with the largest budget in our analysis. The figure visually demonstrates the improvements in demand coverage for different service types and demand groups, reinforcing the effectiveness of the proposed model in enhancing accessibility. 

4. Quality and Consistency of Discussions and Conclusions

o The contributions could be expanded to suggest specific urban planning policies or further research areas directly stemming from the study's findings. 

We have expanded the conclusion to suggest specific urban planning policies that stem directly from the study's findings. We now discuss how the model results can help urban planners identify key areas where zoning changes could promote the inclusion of essential services in underserved areas. This approach would ensure that critical services are more accessible to all population groups. Additionally, we emphasize the importance of enhancing accessibility for vulnerable or mobility-impaired groups by prioritizing investments in accessible infrastructure.

o It would be beneficial if the authors discuss the limitations of their approach, particularly concerning the generalizability of the randomized greedy algorithm across different urban settings. This could offer a more balanced view and suggest areas for further methodological enhancement. 

As mentioned in our previous response, we have added a paragraph in the Conclusions and Future Developments Section, to detail what we believe are the main limitations of our study. We particularly highlight the absence of capacity constraints at facilities, the challenges associated with very large networks, and the demanding data collection process.

o The discussion should more explicitly connect the findings back to the research questions and objectives. Address any unexpected results and discuss their implications.

The discussion in the Conclusion section is expanded to more clearly connect the findings to the study's objectives and discuss their implications. Since this work is prescriptive in nature, it is not driven by specific research questions or hypotheses but rather focuses on providing a decision support tool for urban planners to optimize non-motorized accessibility. To better align the findings with the study's objectives, we have highlighted how the model results help identify key areas where new facilities and accessible roads would most improve accessibility for various demand groups. This directly supports the objective of guiding zoning changes and infrastructure investments that promote inclusive urban development. The revised discussion also emphasizes that the variations observed in solutions for different demand groups suggest targeted policies, such as prioritizing accessible infrastructure development for vulnerable or mobility-impaired populations. These insights underline the practical utility of the model for planners seeking to enhance urban accessibility in a data-driven manner.

D. Minor Comments 

o There are minor typographical and grammatical errors scattered throughout the text that need correction to maintain the professionalism of the manuscript. 

o Correct minor typographical errors, such as: • Page 3: "acounting" should be "accounting" • Page 3: "enahncing" should be "enhancing" • Page 2: "encopasses" should be "encompasses" • Page 3: "heteregenous" should be "heterogeneous" • Page 17: "deeemed" should be "deemed"

Thank you for helping us spotting typos. We have fixed the highlighted errors and a few more we have discovered during the revision process. 

Reviewer 2

The paper entitled Building Non-Motorized Accessible Communities for Heterogeneous Demand: A Facility Location and Network Design Problem is really interesting.

The needs to not only improving public transportation but also non-motorized infrastructure is indeed needed to some extends.

There is concern regarding the English quality on this manuscript as some parts are redundant, especially the background part. Furthermore, there are some parts that should be improved as i found some dangling modifier that reduce the overall paper clarity. Please check again and use English proofreading service to improve the quality and emphasizes the study aims and contributions.

I also notice that the tables & figures presented are somewhat unclear. Rather than using the notation l, n, in the figures or tables, I suggest the authors to use the actual variables. This will ensure better readability.

Please also make one more subsection focusing on managerial implication of the findings.

Thank you for your suggestion. The manuscript has gone through a comprehensive review and proofreading. We made a number of changes in many sections, some of which aimed at addressing policy and managerial implications of our modeling choices and computational results. We have elaborated in Sections 1 and 2 on how the choice of a coverage-based objective aligns with practical urban planning policies, such as the 15-minute city concept. This approach allows policymakers to directly apply the model's results to enhance non-motorized accessibility. Furthermore, in Section 3, the decision to use a randomized greedy algorithm is also linked to possible managerial implications; its simplicity and effectiveness make it easier for urban planners to understand, implement, and communicate results to stakeholders. In Sections 4 and 5, we have expanded the discussion on the policy implications of our model, particularly highlighting how the results can guide urban planners in making zoning changes that promote equitable distribution of essential services. This is directly linked to practical managerial decisions on where to invest in new facilities and accessible infrastructure to maximize accessibility for different demand groups, including vulnerable and mobility-impaired populations.

We used the notation n and l for the figures and tables as the results summarized in both are obtained by creating different budget scenarios changing the values of n and l, in line with the model notation. In the context of these visualizations we don’t believe placing the variable notation (x, y, z) would help digesting the charts and figures. We appreciate and understand the referee’s concern regarding figures and tables readability, so while we believe that the use of l and n, correctly matches the model’s formulation, we are also open to other suggestions or ideas to improve the figures.

---

## [Decision Letter · Decision Letter 1]

4 Oct 2024

Building Non-Motorized Accessible Communities for Heterogeneous Demand: A Facility Location and Network Design Problem

PONE-D-24-23480R1

Dear Dr. Starita,

We’re pleased to inform you that your manuscript has been judged scientifically suitable for publication and will be formally accepted for publication once it meets all outstanding technical requirements.

Kind regards,

Abel C. H. Chen

Academic Editor

PLOS ONE

Additional Editor Comments (optional):

Reviewers' comments:

Reviewer's Responses to Questions

**Comments to the Author**

1. If the authors have adequately addressed your comments raised in a previous round of review and you feel that this manuscript is now acceptable for publication, you may indicate that here to bypass the “Comments to the Author” section, enter your conflict of interest statement in the “Confidential to Editor” section, and submit your "Accept" recommendation.

Reviewer #1: All comments have been addressed

Reviewer #2: All comments have been addressed

2. Is the manuscript technically sound, and do the data support the conclusions?

Reviewer #1: Yes

Reviewer #2: Yes

3. Has the statistical analysis been performed appropriately and rigorously? 

Reviewer #1: Yes

Reviewer #2: Yes

4. Have the authors made all data underlying the findings in their manuscript fully available?

Reviewer #1: Yes

Reviewer #2: Yes

5. Is the manuscript presented in an intelligible fashion and written in standard English?

Reviewer #1: Yes

Reviewer #2: Yes

6. Review Comments to the Author

Reviewer #1: Thank you for addressing my concerns and improve the quality of the manuscript. The current version is acceptable for publication.

Reviewer #2: I have reviewed the authors' responses, and I am satisfied that they have addressed my comments and concerns. However, I recommend that the authors take a more structured approach in their future revisions. It would be helpful if they could clearly indicate the specific location of the changes by providing the section, page numbers, and a clear explanation of how the comments were addressed. This would make it easier for reviewers to locate the changes without having to search through the files.

7. PLOS authors have the option to publish the peer review history of their article (what does this mean?). If published, this will include your full peer review and any attached files.

Reviewer #1: No

Reviewer #2: No

---

## [Editor Report · Acceptance letter]

8 Oct 2024

PONE-D-24-23480R1 

PLOS ONE

Dear Dr. Starita, 

I'm pleased to inform you that your manuscript has been deemed suitable for publication in PLOS ONE. Congratulations! Your manuscript is now being handed over to our production team.

Kind regards, 

on behalf of

Dr. Abel C. H. Chen 

Academic Editor

PLOS ONE